**Data Availability Statement:** All relevant data are in the paper and its Supporting Information files.

# Upper respiratory symptoms (URS) and salivary responses across a season in youth soccer players: A useful and non-invasive approach associated to URS susceptibility and occurrence in young athletes

**Renata Fiedler Lopes[1], Luciele Guerra Minuzzi[1,2], António José Figueiredo[1], Carlos Gonçalves[1], Antonio Tessitore[3], Laura Capranica[3], Ana Maria Teixeira[1], Luis Rama[1]***

1 Research Center for Sport and Physical Activity, University of Coimbra, Faculty of Sport Sciences and Physical Education, Coimbra, Portugal, 2 Exercise and Immunometabolism Research Group, Department of Physical Education, Faculty of Sciences and Technology, São Paulo State University, Presidente Prudente, São Paulo, Brazil, 3 Department of Movement, Human and Health Sciences, University of Rome "Foro Italico", Rome, Italy

* luisrama@fcdef.uc.pt

## Abstract

This study examined the effect of a competitive season on salivary responses [cortisol (sC), testosterone (sT), Testosterone/Cortisol ratio (sT/C), Immunoglobulin A (sIgA), sIgA secretion rate (srIgA), alpha-amylase (sAA)] and upper respiratory symptoms (URS) occurrence in three teams of male soccer players (Under-15, Under-17 and Under-19 yrs.). Training and competition volumes, salivary biomarkers and URS were determined monthly. No differences were found for monthly training volume between teams. Incidence of URS was higher for the U15 (44.9% of the total cases). Higher sT and srIgA were observed for the U19, lower sC were found for the U17 and sAA showed higher values for the U15 throughout the season. In the U15, significant difference ($p = .023$) was found for sIgA concentration with higher concentration values in January compared to December (-42.7%; $p = .008$) and the sT showed seasonal variation ($p < .001$) with the highest value in January significantly different from October (-40.2%; $p = .035$), November (-38.5%; $p = 0.022$) and December (-51.6%; $p = .008$). The U19 presented an increase in sC in March compared to February (-66.1%, $p = .018$), sT/C were higher in February compared to March (-58.1%; $p = .022$) and sAA increased in March compared to September (-20.5%; $p = .037$). Negative correlations, controlled for age group, were found between URS occurrence and srIgA ($r = -0.170$, $p = .001$), sAA ($r = -0.179$, $p = .001$) and sT ($r = -0.107$, $p = .047$). Monitoring salivary biomarkers provides information on mucosal immunity with impact in URS occurrence. Coaches could manipulate training loads to attenuate the physical stressors imposed on athletes, especially at demanding and stressful periods.

**Funding:** LGM are financed by a grant from CAPES – Ministry of Education – Brazil, reference code BEX:1417/13-4. (http://www.capes.gov.br). AMT and LM are registered at Research Center for Sport and Physical Activity - CIDAF (UID/PTD/04213/2016). (https://www.uc.pt/fcdef/Investigacao/in_english/CIDAF). The funders had no role in study design, data collection and analysis, decision to publish, or preparation of the manuscript.

**Competing interests:** The authors have declared that no competing interests exist.

# 1 Introduction

The long competitive season often includes a high frequency, intensity and duration of training sessions, that can place a heavy strain on biological systems of young players. Furthermore, the increase of exercise intensity, occurring as part of the success paradigm towards elite sport, could be also added to a psychophysiological stress on youth elite players [1].

Long-term hormonal responses to elite team sports have been reported [2–6]. In particular, sC, sT and their sT/C values are considered markers of exercise-related stress and balance between anabolic and catabolic processes, respectively [5,7,8]. Depending on the intensity and duration of exercise, transient increments or suppressions in immune parameters may also occur, namely those involved in mucosal immunity including sAA [9] and sIgA [8,10].

Produced locally in the salivary glands and controlled by the autonomous nervous system, sAA inhibits bacterial adherence and growth to epithelial surfaces [11] and has been proposed as a biomarker of body stress and sympathetic nervous system activity [12–14]. Preventing the attachment of external pathogens to mucosal surfaces, acting as a first line of defence against microbial invasion, secretory IgA also plays an important role in mucosal immunity [12,15,16], which might be affected by strenuous bouts of intense exercise [17]. In particular, multiple daily intense training sessions may result in accumulative mucosal immune suppression of sIgA and srIgA [15,16].

Exercise-induced transient suppression of immunological responses could determine an impaired immune condition [17] increasing the risk of contracting upper respiratory tract infections (URTI) [17–23]. The diagnosis of URTI is still unclear, especially in the absence of laboratory testing confirming the presence of a pathogen [17,24]. Indeed, the term upper respiratory symptoms (URS) could be more appropriate to classify signs and symptoms affecting the upper airways [24–27].

Despite the many studies on various acute immunological and endocrine responses to training and competition [20,28–30], data from seasonal monitoring of biomarkers using less invasive strategies as saliva collection is still lacking [31]. In this study, we will permit a better understanding of the load dynamics in the last three age-groups of the youth department and, as well, the age-groups where the game has is formal format (eleven vs. eleven). With this in mind it's possible to the coaches and researchers to highlight the mechanisms that are underpinning the training sessions and, by inference, the competition, putting all in line with the progressive demands of soccer talent development programs. Monitoring young athletes can provide valuable information to coaches about the adequacy of their training plans; the aim of this study was to examine the monthly variations of immune and hormonal mucosal parameters in young soccer players during a competitive season. It has been hypothesized that: 1) youth soccer players could show different age-related adaptive hormonal and immune responses to training and competition; 2) exercise could mediate changes in mucosal immune parameters and the incidence of upper respiratory symptoms could be related.

# 2 Materials and methods

## 2.1 Subjects

Fifty-seven male young players from the same soccer club academy, playing at national level in their age group (main Portugal national championship), were recruited for this study. At the beginning of the study U15, U17 and U19 teams encompassed 28, 35 and 28 players respectively; however, some dropouts (11, 13 and 10 soccer players, respectively) occurred during the competitive season due to: the club selection program requirements, occasional maladaptation's, injury and other reasons. Thus, the final sample consisted of the seventeen Under 15

(U15: age 14.8±0.2 yrs.; stature 170.8± 6.6 cm; body mass 60.1±9.5 kg; body fat 14.8±3.8%), twenty-two Under 17 (U17, mean ± SD: age 16.1±0.4 yrs.; stature 174.4±6.7 cm; body mass 67.0±8.1 kg; body fat 14.1±3.0%) and eighteen Under 19 (U19, mean ± SD: age 18.8±0.2 yrs.; stature 177.5±6.9 cm; body mass 70.9±8.1 kg; body fat 13.3±4.9%). The recruitment of soccer players was agreed with the technical department of the soccer academy at the end of the season that preceded the study (May/June), and the study included all players from the squads of the various teams. The inclusion criteria were: a) to volunteer, as well as to authorize their tutors to participate in the study; a) integrate the team of the respective age group up to one month after the start of the season; b) do not present any physical limitation or injury that prevents them from participating in the regular training of the team. Exclusion criteria: a) the occurrence of an injury or illness that would require you to leave regular training for a period longer than 2 weeks; b) failure to participate in at least two thirds of the training plan; c) transfer to another club / team. The sample consists of players who participate in the main national competition of the respective age groups, obtaining merit sports results. The study took place in a training academy for football players belonging to one of the historic clubs of Portuguese football. The study was fully approved by the Ethics and Human Subjects Review Board of the Faculty of Sports Science and Physical Education of the University of Coimbra (Portugal) and conducted according to the Helsinki declaration (CE/FCDEF-UC/00032013). Verbal and written information on the experimental procedures were provided for all participants. Signed informed consent statements were obtained from athletes and their parents or guardians when underaged (under 16).

### 2.2 Design

The study followed a longitudinal parallel design. To verify different age-related adaptive responses to training, salivary hormonal (i.e., sC, sT and sT/C) and immune parameters (i.e., sAA, sIgA, srIgA) were monitored during a competitive season spanning from July to April. The relationship between mucosal immune parameters and the incidence of URS was verified monthly, always in the first training session in the first week of each month, after two days of recovery for all teams (U15 = 8 months; U17 and U19 = 9 months).

### 2.3 Training and competition load

The training load was computed by counting the time (minutes) spent in training sessions and competitions. During the experimental period, the three groups had four (U15 and U17) and five (U19) training sessions with 1.5h of duration/session plus 70-, 80- and 90-minute match per week for U15, U17 and U19, respectively. The typical training sessions consisted of warm-up, individual and group technical drills, team technical drills, circuit training and cool-down.

### 2.4 Procedures and data collection

During the study period, no dietary interventions were undertaken. Players were instructed to maintain their normal daily nutritional and hydration intake. Saliva samples were collected monthly before the first training session of the week, ensuring at least 36 hours of rest from the last match or training, which enabled the recovery of acute immune and hormonal responses determined from previous exercise [32]. Furthermore, subjects were asked to avoid any intense exercise during the 36 hours before each experimental session and to abstain from food and caffeine products intake two hours before to saliva collection. Once at the collection site, subjects were required to rinse out their mouths with distilled water to clean the oral cavity 20 min before collection time (18.00–19.00 pm). Then, passive unstimulated whole saliva samples were collected during 3 min with athletes in a seated position and with the head tilted slightly

forward, using appropriate pre-weighted and pre-labeled plastic tubes of 7mL (Sarstedt®) and immediately stored at approximately 4˚C in a polystyrene container with ice, prior to transport and subsequent storage in the laboratory. Within the testing laboratory, the samples were de-identified by application of a laboratory number, weighted again to calculate saliva flow rate and they were thawed, vortexed, and centrifuged at 1,500 g (@3,000 rpm) for 15 min as per ELISA protocol (Salimetrics, Carlsbad, CA, USA) and split into two Eppendorf tubes to allow assay re-run if required. Labeled saliva samples were frozen at −80˚C within a secure, back-up powered ultra-low freezer within 4 h of collection. No additional preservatives such as sodium azide were added to the samples to exclude possible assay interference.

## 2.5 Salivary hormonal and immune assessments

Concentrations of sC, sT, sAA and sIgA were determined using commercially available ELISA kits (*Salimetrics*, Inc., State College, PA, USA). The concentration of salivary IgA was expressed in term of: 1) the absolute concentration of salivary IgA ($\mu g \cdot ml^{-1}$) and 2) the salivary IgA secretion rate ($\mu g \cdot min^{-1}$). The salivary IgA secretion rate was calculated by multiplying the absolute salivary IgA concentration ($\mu g \cdot ml^{-1}$) by the salivary flow rate ($ml \cdot min^{-1}$); this latter value was calculated by dividing the total volume of each saliva sample (ml) by the time taken to collect each sample (3 min). Salivary secretion rate was calculated from saliva flow rate ($ml.min^{-1}$), which was determined dividing the saliva volume by the collection time. Saliva flow rate of valid samples should not be <0.1 $ml.min^{-1}$. Under basal conditions, the rate of saliva production is 0.5 $ml.min^{-1}$ [33]. All samples belonging to the same athlete were tested in the same plate to reduce inter-assay variations and in duplicate. The intra-assay maximum coefficient of variation was 6.7% for sAA, 3.65% for sC, 3.3% for sT and 3.3% for sIgA. The inter-assay maximum coefficient of variation was 5.8% for sAA, 6.41% for sC, 8.1% for sT and 7.9% for sIgA.

## 2.6 Monitoring of URS episodes

Subjects were required to fill a monthly log to document any signs or symptoms of URS including cold, cough, nasal secretion, headache, sore throat, muscle pain, diarrhoea, abdominal pain, cold shivers, itchy eyes, sneezing and fever. The athletes' information was compared with that provided from their respective coaches, to eventually confirm the reported URS occurrence. When a sign of illness was reported the WURSS-21 [34] was applied. This questionnaire was developed to comprehensively measure all significant health-related dimensions that are negatively affected by the common cold. A conservative method of identifying URS was used. The logging of an episode required a report of two or more cold-specific symptoms during at least two days in a row. A new episode was considered after a minimum interval of 10 days following the previous one [35]. Symptoms related to allergic episodes (itchy eyes, sneezing), gastrointestinal or muscular pain related to injuries were carefully analysed and discarded [24,27].

## 2.7 Statistical analysis

Descriptive statistics was computed as mean and standard deviation (mean ± SD). The Chi-square test was used to compare time spent in training and competition and URS occurrence of the 3 soccer groups. Accounting for non-normal distribution and small sample size the Friedman test analysed the within repeated measures and Kruskal-Wallis test was used for between comparisons. Pairwise multiple comparisons were conducted with the adjustment by Dunn test originally designed first for Kruskal-Wallis test, able to use in non-parametric repeated measures and incorporated in SPSS 21.0. Effect size was used to ascertain magnitude

of the difference of the mean as trivial (0–0.19), small (0.20–0.49), medium (0.50–0.79), large (0.80 and greater) [36]. We calculated the a priori statistical power. For an α = 0.05 and moderate effect size (*es* = 0.5), the sample size of our study assures an β of 0.84 (84%) using G\*Power Version 3.1.9.2. The exploration of association between biomarkers and URS was done through Spearman Rho (p≤ 0.05). Statistical analysis was performed using the software package (IBM SPSS, version 21.0, 2012), with statistical significance set at (p≤ 0.05).

## 3 Results

### 3.1 Training load

Regarding the monthly distribution of exercise volume in each group, the U15 highest percentages emerged in October (12.3%), whereas the relative picture for both U17 (18.6%) and U19 (14.5%) was in August (above each group seasonal training volume average). This distribution fits with the previous pre-season overload of both U17 and U19 and mimics the incremental approach planned for the youngest group (U15), which performs a higher volume later. Table 1 shows detailed data of the monthly exercise volume (minutes) spent in training and matches. A seasonal training volume variation for all groups are presented in the Fig 1.

### 3.2 URS occurrence

Higher URS occurrence emerged for U15 team (40 episodes,44.9% of the total) compared to U17 (33 episodes, 37.1% of total) and U19 (16 episodes, 18% of the total) (Z = 6.47; P = 0.039; Fig 2). A significant inverse association was found between URS with srIgA (r = -0.170, p = .001), sAA (r = -0.179, p = .001) and sT (r = -0.107, p = .047). Furthermore, the correlation between the number of URS events and the age group (r = -0.180, *p*< .001) showed that there was a trend to younger players were more prone to get an URS episode.

### 3.3 Seasonal variation of salivary biomarkers

Table 2 reports the comparative analysis of the immunological and endocrine biomarkers values (mean and standard deviation) and the groups comparisons, whereas Fig 3 (Graphs A-F) show the pattern across time for the physiological parameters, which highlights the most critical variation of the study variables during the soccer season in the three soccer teams.

**Table 1. Monthly training volume and matches of each soccer team (U15, U17 and U19).**

|  | Under-15 | | | Under-17 | | | Under-19 | | |
|---|---|---|---|---|---|---|---|---|---|
|  | Training volume | Match volume | Total volume | Training volume | Match volume | Total volume | Training volume | Match volume | Total volume |
|  | min | min | min | min | min | min | min | min | min |
| August | 1440 | 280 | 1720 | 1730 | 450 | 2180 | 1820 | 360 | 2180 |
| September | 1530 | 280 | 1810 | 1500 | 270 | 1770 | 1530 | 450 | 1980 |
| October | 1620 | 280 | 1900 | 1660 | 270 | 1930 | 1805 | 270 | 2075 |
| November | 1080 | 280 | 1360 | 1520 | 360 | 1880 | 1455 | 270 | 1725 |
| December | 1440 | 210 | 1650 | 1270 | 360 | 1630 | 1385 | 270 | 1655 |
| January | 1350 | 280 | 1630 | 1760 | 270 | 2030 | 1740 | 360 | 2100 |
| February | 1530 | 280 | 1810 | 1455 | 270 | 1725 | 1440 | 180 | 1620 |
| March | 1440 | 210 | 1950 | 1235 | 360 | 1595 | 1355 | 540 | 1895 |
| April | - | - | - | 1445 | 360 | 1805 | 1365 | 360 | 1725 |
| *Mean* | *1428,8* | *262,5* | *1691,3* | *1508,3* | *330,0* | *1838,3* | *1566,3* | *340,0* | *1903,8* |
| *SD* | *162,7* | *32,4* | *164,5* | *185,1* | *63,6* | *188,7* | *192,2* | *115,4* | *215,3* |
| *CV* | *11,4* | *12,3* | *9,7* | *12,3* | *19,3* | *10,3* | *12,3* | *33,9* | *11,3* |

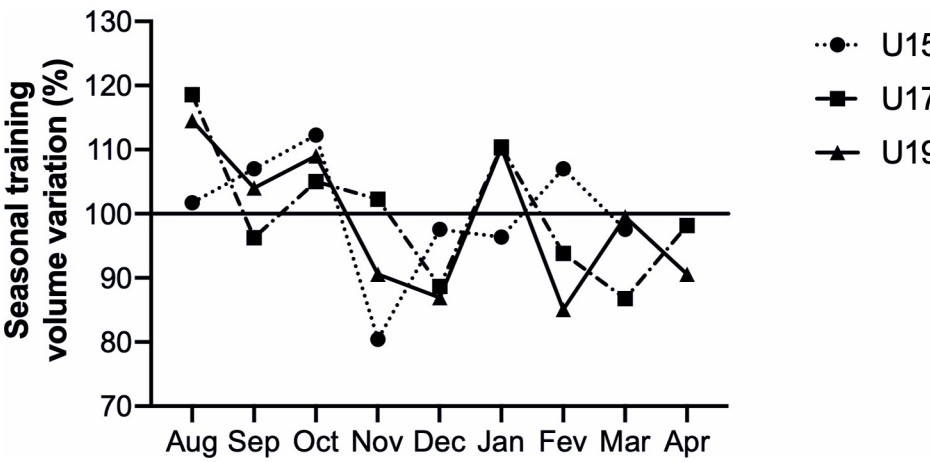

**Fig 1. Seasonal training volume variation of young soccer players (U19, U17 and U15) across a soccer season.** Data are % variation per group.

**3.3.1 Salivary IgA(sIgA) and secretion rate (srIgA).** Significant difference ($\chi^2 = 16.200$; $p = .023$) was found for sIgA concentration in U15 team with higher concentration values in January compared to December ($p = .008$, ES = 1.03 [0.25, 1.8]) (Table 2). Inter-group sIgA comparisons showed only difference between U15 and U17 in January ($p = .016$, ES = 0.92 [0.25, 1.58]), whereas no seasonal differences emerged for sIgA and srIgA between U17 and U19 (Table 2, Fig 3A and 3B). Across the season, higher srIgA values were found in the U19 compared to the U17 in August ($p = .005$, ES = 1.12 [0.45, 1.79]), September ($p = .001$, ES = 1.23 [0.55, 1.90]) and April ($p = .011$, ES = 0.86 [0.21, 1.51]) (Table 2, Fig 3B). Additionally, the U19 showed higher srIgA values with respect to U15 in November ($p = .018$, ES = 1.21 [0.49, 1.93]) (Table 2, Fig 3B).

**3.3.2 Salivary alpha-amylase (sAA).** The U15 and U17 groups showed a stable response for sAA along the season, while the U19 presented significant variation ($\chi^2 = 16.711$; $p = .033$) with the values found in March (by the end of the season) clearly lower than from those registered in September ($p = .037$, ES = $^-$0.48 [-1.15, 0.18]) (Table 2). Differences between U15 and U17 were observed in September ($p = .011$, ES = 1.02 [0.34, 1.69]), October ($p = .014$, ES = 0.80 [0.14, 1.45]), December ($p < .001$, ES = 1.59 [0.87, 2.32]) and March ($p = .002$, ES = 1.86 [0.50,

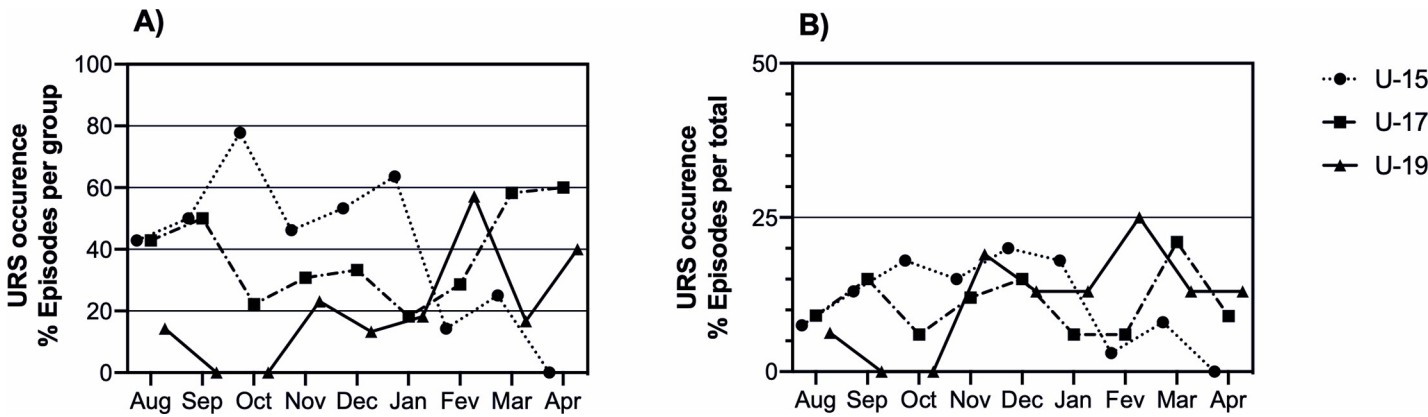

**Fig 2. Upper respiratory symptoms occurrence in young soccer players (U19, U17 and U15) across a soccer season.** Data are: A) URS occurrence of number of episodes in each group per month; B) URS occurrence of % total number of episodes of total group per month.

**Table 2. Salivary immune and hormonal responses (S-IgA, Sr-IgA, S-AA, S-C, S-T and S-T:C ratio) of young soccer players (U19, U17 and U15) across a soccer season.**

| | Aug | Sep | Oct | Nov | Dec | Jan | Feb | Mar | Apr |
|---|---|---|---|---|---|---|---|---|---|
| | | | | **Under 15 (n = 17)** | | | | | |
| S-IgA (mg.dL$^{-1}$) | 279.3 (97.1) | 302.1 (95.2) | 285.6 (70.0) | 240.7 (124.9) | 224.3 (132.4) | 391.2(189.7)[e] | 248.6 (101.6) | 280.5 (136.8) | |
| sr-IgA (µg.min$^{-1}$) | 98.5 (54.9) | 85.4 (31.6) | 78.0 (30.2) | 72.1 (26.2) | 84.5 (58.2) | 108.5 (59.5) | 65.6 (34.5)[f] | 74.9 (33.4) | |
| S-AA (U.mL$^{-1}$) | 95.6 (60.5) | 87.1 (45.9) | 97.8 (49.1) | 81.1 (40.4) | 115.3 (57.0) | 109.6 (109.9) | 116.2 (72.8) | 93.9 (59.1) | |
| S-C (g/mL) | 208.7 (82.4) | 195.3 (77.9) | 132.4 (82.4)[a] | 144.5 (124.8) | 124.3 (98.1) | 119.6 (48.7) | 154.3 (43.0) | 165.6 (119.9) | |
| S-T (ng/mL) | 87.3 (34.5) | 69.6 (30.2) | 65.9 (29.6) | 67.8 (21.6) | 53.3 (18.0) | 110.2 (60.2)[c,d,e] | 72.9 (15.5) | 65.6 (27.9) | |
| S-T:C | 0.49 (0.28) | 0.45 (0.37) | 0.62 (0.32) | 0.69 (0.31) | 0.74 (0.62) | 1.25 (1.21) | 0.51 (0.20) | 0.49 (0.24)[cf] | |
| | | | | **Under 17 (n = 22)** | | | | | |
| S-IgA (mg.dL$^{-1}$) | 368.3 (229.5) | 239.6 (117.9) | 289.8 (151.4) | 256.8 (153.1) | 270.3 (146.2) | 245.9 (128.4) £ | 293.8 (134.4) | 247.9 (117.5) | 227.0 (96.5) |
| sr-IgA (µg.min$^{-1}$) | 68.5 (41.1) | 69.0 (41.6) | 72.5 (53.0) | 99.8 (58.1) | 120.8 (107.6) | 89.2 (45.5) | 121.2 (102.6) | 91.2 (90.3) | 63.1 (41.8) |
| S-AA (U.mL$^{-1}$) | 54.5 (42.4) | 45.6 (43.9)£ | 54.4 (57.6)£ | 67.6 (99.5) | 41.2 (36.3)£ | 55.5 (60.1) | 58.3 (65.6) | 33.9 (43.0)£ | 47.3 (36.5) |
| S-C (ng/mL) | 130.5 (107.8)£ | 85.8 (63.0)£ | 56.0 (39.6)£ | 66.3 (54.7)£ | 49.1 (34.0)£ | 83.2 (41.6) | 84.1 (45.6)£ | 119.3 (77.0) | 101.1 (78.1) |
| S-T (ng/mL) | 84.8 (21.0) | 65.5 (19.0) | 71.5 (26.6) | 64.8 (21.8) | 60.5 (17.0) | 75.3 (27.0)£ | 107.3 (64.5) | 80.1 (28.3) | 64.0 (20.4) |
| S-T:C | 1.17 (1.0) £ | 1.14 (0.94)£ | 1.57 (0.87)[g,h] £ | 1.22 (0.56) £ | 1.80 (1.16)£ | 1.36 (1.53) | 1.40 (0.63) £ | 1.13 (1.05) | 1.10 (0.92) |
| | | | | **Under 19 (n = 18)** | | | | | |
| S-IgA (mg.dL$^{-1}$) | 352.2 (111.2) | 241.8 (66.8) | 287.4 (145.5) | 277.9 (137.7) | 281.2 (128.5) | 290.9 (85.3) | 251.8 (126.3) | 286.4 (116.1) | 251.5 (91.5) |
| Sr-IgA(µg.min$^{-1}$) | 154.3 (104.3)¥ | 145.5 (80.6)¥ | 117.8 (103.9) | 153.6 (90.1)£ | 170.6 (136.0) | 150.3 (98.5) | 120.0 (95.5) | 103.7 (57.2) | 118.0 (83.1)¥ |
| S-AA (U.mL$^{-1}$) | 54.9 (51.6) | 56.2 (47.7) | 74.2 (73.6) | 91.3 (99.4) | 61.1 (44.8)£ | 71.2 (42.8) | 54.9 (52.7) | 37.5 (27.0) [b] | 60.0 (46.9) |
| S-C (ng/mL) | 232.5 (175.9) | 243.9 (163.0)¥ | 226.1(133.3)¥ | 179.1 (138.2)¥ | 236.4 (175.2)¥ | 222.9 (126.5)¥ | 99.3 (50.4)¥ | 292.7 (191.2) [g,] ¥ | 256.0 (210.2)¥ |
| S-T (ng/mL) | 90.5 (30.7) | 95.4 (20.9)¥£ | 87.7 (19.1) | 82.1 (21.4) | 90.8 (29.6)¥,£ | 87.9 (21.3) | 121.4 (39.0)£ | 97.9 (36.3)£ | 98.4 (33.9)¥ |
| S-T:C | 0.68 (0.69) | 0.73 (0.67) | 0.64 (0.63) ¥ | 0.92 (0.91) ¥ | 0.68 (0.60) ¥ | 0.66 (0.72)¥ | 1.36 (0.46) [h]£ | 0.57 (0.67)¥ | 0.70 (0.69) |

Data are mean (SD).

[a] significant difference with August

[b] significant differences with September

[c] significant difference with October

[d] significant difference with November

[e] significant difference with December

[f] significant difference with January

[g] significant difference with February

[h] significant difference with March

[i] significant difference with April, (p<0.05).

¥ significant difference with U-17

£ significant difference with U-15, p<0.05.

1.87]) whilst differences between U15 and U19 were observed in December ($p$ = .034, ES = 1.06 [0.35,1.77]) (Table 2, Fig 3C).

**3.3.3 Salivary cortisol (sC).** The U15 showed a significant variation during the season ($\chi^2$ = 15.309, $p$ = .032) with higher values early in the season (August compared to October) ($p$ = .032, ES = 0.93 [0.2, 1.63]). No differences were observed for sC in the U17 group along the season. The U19 group also exhibit significant variation during the season ($\chi^2$ = 16.133; $p$ = .041) reaching higher values in March compared to February ($p$ = .018, ES = 1.32 [-2.15, -0.48]) (Table 2).

Comparing the team's data along the season, the U17 showed lower sC than the U19 and U15 groups ($p$ < .001) (Table 2, Fig 3D). Looking at the monthly variation, U17 compared to U19 registered lower concentrations of sC in September ($p$ = .001, ES = 1.33 [0.64, 2.02]), October ($p$ < .001, ES = 1.81 [1.07, 2.55]), November ($p$ = .012, ES = 0.85 [0.19, 1.51]),

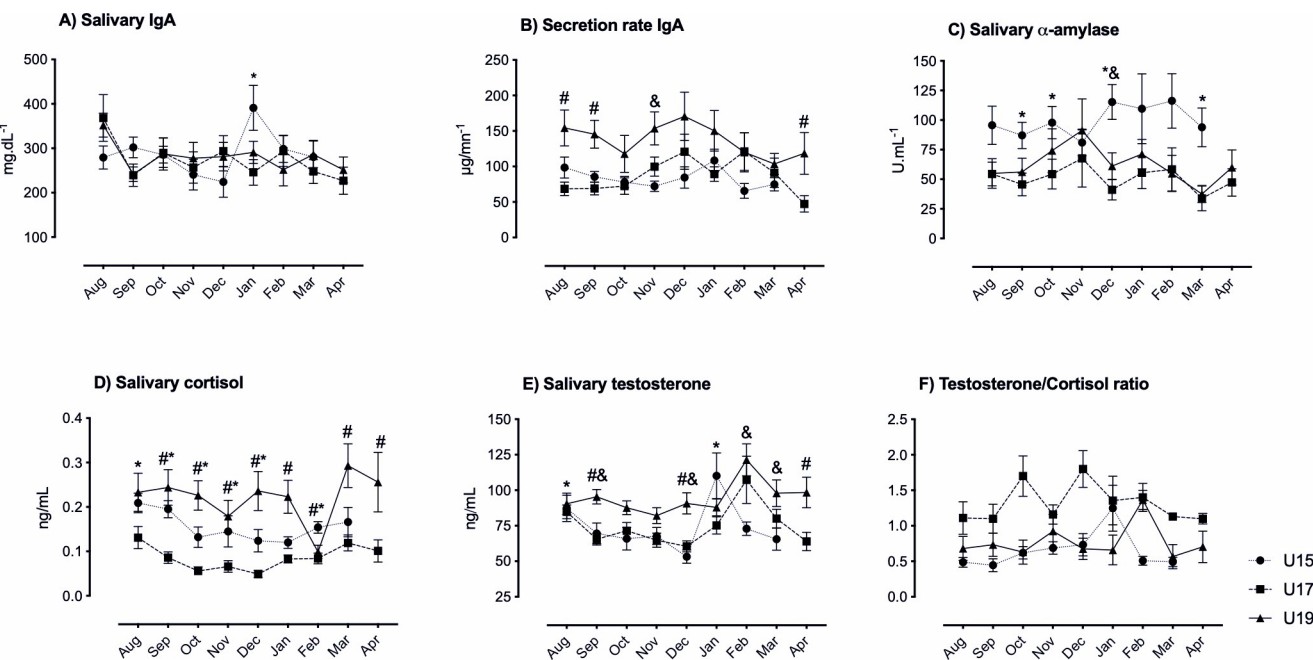

**Fig 3. Salivary immune and hormonal responses of young soccer players (U19, U17 and U15) across a soccer season.** Data are Mean + SD (standard deviation). * P < 0.05 between U15 and U17. # P < 0.05 between U17 and U19. & P < 0.05 between U15 and U19. Abbreviations: U15: Under-15 yrs, U17: Under-17 yrs, U19: Under-19 yrs.

December ($p <$ .001, ES = 1.56 [0.85, 2.27]), January ($p =$ .002, ES = 1.55 [0.84, 2.26]), February ($p =$ .008, ES = 0.32 [-0.31, 0.94]), March ($p =$ .007, ES = 1.23 [0.55, 1.97]) and April ($p =$ .019, ES = 1.01 [0.35, 1.68]) (Table 2, Fig 3D). Compared to U15, U17 had lower concentrations in August ($p =$ .003, ES = 0.8 [0.14, 1.45]), September ($p =$ .003, ES = 1.56 [0.84, 2.29]), October ($p =$ .012, ES = 1.23 [0.54, 1.92]), November ($p =$ .038, ES = 1.11 [0.44, 1.78]), December ($p =$ .011, ES = 1.08 [0.41, 1.76]) and February ($p =$ .001, ES = 0.54 [0.10, 1.18]) (Table 2, Fig 3D). No differences were found for the cortisol concentration between the U15 and U19 groups (Table 2, Fig 3D).

**3.3.4 Salivary testosterone (sT).** In the U15 the sT showed seasonal variation ($\chi^2 =$ 26.667; $p <$ .001) with the highest value in January, that was significantly different from October ($p =$ .035, ES = 0.93 [0.23, 1.64]), November ($p =$ 0.022, ES = 0.94 [0.23, 1.65]) and December ($p =$ .008, ES = 1.28 [0.54, 2.02]) (Table 2). Furthermore, differences in sT concentrations were found between U15 and U19 in September ($p =$ .001, ES = 0.99 [0.3, 1.7]), December ($p <$ .001, ES = 1.52 [0.76, 2.27]), February ($p =$ .002, ES = 1.61 [0.85, 2.38]) and March ($p =$ .039, ES = 0.99 [0.29, 1.70]) (Table 2, Fig 3E). Whilst sT differences between U15 and U17 teams were observed only in January ($p =$ .0016, ES = 0.78 [0.13, 1.44]) (Table 2, Fig 3E). The sT concentrations in the U17 and U19 teams differed significantly in September ($p =$ .002, ES = 1.5 [0.8, 2.21]), December ($p =$ .004, ES = 1.29 [0.60, 1.94]) and April ($p =$ .005, ES = 1.26 [0.58, 1.94]) (Table 2, Fig 3E).

**3.3.5 Salivary testosterone to cortisol ratio (sT/C).** The U15 team showed significant variation in this anabolic:catabolic ratio ($\chi2 =$ 24.470; $p =$ .002) that was lower in March compared to October ($p =$ .035, ES = 0.48 [-0.20, 1.16]) and January ($p =$ .001, ES = 0.88 [0.18, 1.59]) (Table 2). The U17 team showed differences over the season ($\chi2 =$ 21.504; $p =$ .006) with higher values in October when compared to the end of the season in March ($p =$ .037, ES = 0.29 [-0.3, 0.89]) and April ($p =$ .037, ES = -0.48 [-1.08, 0.12]) (Table 2). The U19 team

also showed a significant variation ($\chi2$ = 17.128; $p$ = .029) with the highest values in February when compared to those observed in March ($p$ = .022, ES = -1.37 [0.65, 2.1]) (Table 2).

Several differences were found between groups in the sT/C (Table 2, Fig 3F). The U17 team had higher values than U19 team in October ($p$ < .001, ES = 1.01 [0.34, 1.67]), November ($p$ = .030, ES = -0.3 [-0.94, 0.31]), December ($p$<0.001, ES = -1.8 [-1.85,-0.3]), January ($p$ = -.008, ES = -0.57 [-1.2, 0.06]) and March ($p$ = .003, ES = -0.62 [-1.26, 0.02]); when compared to the U15 team, higher values were found in August ($p$ = .046, ES = 0.8 [0.14, 1.46]), September ($p$ < .001, ES = 0.87 [0.21, 1.53]), October ($p$ = .005, ES = 1.08 [0.40, 1.75]), November ($p$ = .046, ES = 0.91 [0.25, 1.57]), December ($p$ < .001, ES = 1.1 [0.42, 1.78]) and February ($p$ = .001, ES = 1.81 [1.06, 2.56]) (Table 2, Fig 3F). Between the U19 and U15 teams, increased values in the older group were only found in February ($p$ = .001, ES = 2.37 [1.51, 3.24]) (Table 2, Fig 3F)

## 3.4 Training load and salivary biomarkers

The total volume of training showed a positive correlation with mean sIgA concentrations (r = 0.104, $p$ = .036), whereas the total volume of competitions correlated negatively with sAA enzyme activity (r = -0.138, $p$ = .006).

## 4 Discussion

To our knowledge, this is the first study that monitored the occurrence of URS, the immune and hormonal responses during an entire soccer season in relation to the training and competition load of three age-groups of youth players belonging to the same soccer academy. Main findings highlighted tendencies to associations between training volume, salivary biomarkers and URS occurrence.

### 4.1 Seasonal training load variation

We observed a stable training volume in each group across de season (Table 1). The CV of the total volume was 9,7%, 10,3% and 11.3% for U-15, U-17 and U-19 respectively. These CV was lower than that observed in Portuguese professional soccer players [37]. Valente-dos-Santos and co-authors [38] showed that the overall training demand in soccer tends to stabilize after the player's peak height velocity.

### 4.2 Immune and hormonal responses along the training season

**4.2.1 Salivary immune parameters.** Considering all collected immune data over the season in the three age-group teams, we observed that the U19 team tended to show higher mean srIgA values than the younger teams and the opposite behavior for the sAA mean concentration.

Looking for the seasonal variation in each age-group, in the U15 team, sIgA showed the lowest value in December. This is in line with the observed pattern in rugby players during an eleven months season [31], where lower IgA levels were found in December and explained by a period of increased training intensity and reduced match activity, the same as was observed in U15 team. Although not statistically significant, the U19 team also showed a trend toward a decrease in srIgA in the last three months of the season. This behavior could be linked to the cumulative impact of a phase of intense training load and the cold weather conditions at this time of the year in the Northern hemisphere. Similarly, a study showed that sIgA had a significant decline in professional English Premier League soccer players during an intensive winter training period [39]. These results have practical relevance because they demonstrate that sIgA provides a non-invasive assessment that is simultaneously sensitive to the changes to either

physical and/or psychological stress [28] associated with a winter short-term period of intensive competition.

U19 showed marked sAA differences along the training season (Mar *vs* Sep). It is possible to speculate that sAA significantly changes due to physiological and psychological stress in acute conditions [38]. In endurance sports, sAA acts as a potential marker of training overload [40] and adaptation [41], but in team sports the prevailing idea is that sAA is probably not very responsive and not very useful for longitudinal monitoring of the immune response. In fact, a prolonged period of training and the participation in a high number of matches could lead to a lower anti-microbial defense, affecting the immune response of young players. This could be seen in the U19 teams that showed a tendency of decrease in sAA values in March, the most stressful competitive month for this team.

**4.2.2 Salivary hormonal parameters.**   The current study highlighted that U19 increased sC values in March and in April, which could be explained by the participation to decisive matches. In fact, training records showed that March was highly demanding, with a sequence of nine matches, being all decisive for the main goal of the season. The stress was also elevated at the end of the season, when match outcomes were important to maintain the U19 in the first division of the national championship, players were also stressed with that decisive phase. The higher sC values found in U19 compared to U17 and U15 could also be due to the higher competitive demands of their championship.

The U15 team showed higher sT levels and sT/C ratio in January compared to August. It could be possible to speculate that the rest days during the Christmas vacations helped the recovery process and consequently the immune and hormonal systems, allowing an increase in the anabolic response. The antagonist role of the stress hormone cortisol on mucosal immunity [31] was not confirmed in this study. The scarce variation observed throughout the season for the sC levels did not seem to contribute to the impairment in mucosal immunity. Probably, this group of young soccer players responded well to the stress situations related to training and competition. In general, a tendency for the stabilization of the hormonal system in the U15 was observed. Among the teams, U17 presented small variations in the studied biomarkers. The hormonal response of these young soccer players seems to be well adapted to their regular training workload and matches, one probability could be the "natural selection" process and for this reason was more resilient with respect to their youngest counterparts. It could be hypothesized that players more prone to illness could have dropped out soccer or were not selected for the U17 team. As expected, the U19 tended to show higher sT concentrations compared to the younger groups throughout the competitive season.

The usefulness of the individual hormonal response in team sport athletes for an early detection of maladaptation's over extended periods is controversial. The apparent antagonist role of cortisol and testosterone has already been discussed [11,28]. Results from longitudinal studies have been equivocal and included a rise in sC during a soccer season [6], whereas a reduction in sC and an increase in sT/C were observed during 12 weeks of Australian Football season [2]. Low sC values have been associated to a good recovery, indicating the best moment to increase the training intensity [42]. Also, increases in the testosterone concentration reflect a good recovery of the athletes, thus enabling an increment in training load [43,44]. Conversely, high cortisol levels have been related to the exhaustion of adrenal glands, associated to extreme fatigue [45].

## 4.3 Immune and Hormonal changes and URS occurrence

The demands of modern elite soccer, where the weekly microcycle presents several training units and often more than one competition, make players more likely to experience repeated stress situations with a limited opportunity to recover, resulting in decreases in sIgA

concentrations and increases in the risk for URTIs occurrence [46]. Youth elite soccer players undergo weekly training plans that mimic those of adult elite players. In fact, soccer clubs support early specialization of potential elite players to have, as soon as possible, important revenues [47]. The tendency of inverse association of srIgA levels and URS occurrence between the volumes of training observed in this study reinforces the concept that youth athletes should be exposed to a training load adequate to their age [1]. In fact, the higher occurrence of URS in the U15 players compared to their older counterparts could reflect the difficulties in the adaptation of the immune system related to training during their competitive year. Conversely, players older than 15 years might have a stronger immune response possibly because of a natural sports selection process, which retains the more resilient athletes. In Portugal the highest seasonal prevalence of URS occurs in January [48]. In the current season, the epidemic flu period occurred between the last week of December and the end of February, with the weekly highest incidence rate observed in the end of December. This could explain the incidence rate our study, namely regarding December. In December, more than 50% of the soccer players showed a concurrent depression in their immunity with low salivary IgA concentrations. This inverse relationship between IgA concentration and cases of URS has already been reported in literature [18,20,28,31]. In the U17 players, 20% of the athletes showed symptoms of respiratory illness in September, despite the lower accumulation of training and the low incidence of respiratory infection symptoms in the Portuguese population. In the Under-19 players, the pattern was closer to the one database [48]. Despite the high incidence of cold cases in the Portuguese population in January, regular intense soccer training could be a co-factor that helped explain the increase in URS in these athletes. Finally, the higher URS occurrence in youth players with respect to their older counterparts could be associated with periods of increased stress and training accumulation, resulting independent from environmental conditions. It is likely that young athletes more prone to illness could have dropped out of soccer or have failed the selection for the following championships.

## 5 Conclusions

The current study confirmed that youth soccer players show different age-related adaptive hormonal and immune responses to training and competition. There is an association between training load (volume), mucosal immunity, hormone levels and URS. The monitoring of salivary biomarkers is a noninvasive strategy that identified periods when decreases in salivary IgA and testosterone indicate impaired health, which in turn could affect performance mainly in younger athletes at the beginning of a serious competitive career. A stimulating effect of testosterone on sIgA and of cortisol on sAA was observed, also confirming that mucosal immune parameters and incidence of upper respiratory symptoms are related. The monitoring of sIgA, testosterone and α-amylase levels could provide a useful and non-invasive approach associated to URS susceptibility and occurrence in young athletes during an entire soccer season, helping detect those more prone to illness. With this knowledge, coaches can potentially manipulate daily training loads to attenuate the physical stressors imposed upon the athletes and thereby decrease the likelihood of URS occurrence, especially at more demanding and stressful periods. In considering that in youth soccer players the prevalence of URS is higher with respect to the seasonal influence of severe weather conditions, coaches are urged to consider the effect of accumulation of training load and matches on the athletes' immune response.

### Limitations of this study

In this study some limitations must be acknowledged. First, the quantification of training load was limited to the volume of training and time spent in competition, not considering training

intensity that could have an impact on the immune and hormonal salivary markers [28]. Secondly, the impact of blood contamination on the measurement of salivary hormones, associated with a minor injury in the oral cavity was not examined, as we did not use any biomarker for blood contamination. However, we believe that we adopted trustable procedures. All subjects in the study were checked on a regular base for oral health by the academy medical staff. Additionally, when we detected some values out of the range, they were excluded from the analysis.

S1

## Supporting information

**S1 Data.**
(XLSX)

## Acknowledgments

Firstly, the authors would like to thank all the athletes that volunteered to participate in this study, and the support from the Faculty of Sport Sciences and Physical Education, University of Coimbra.

## Author Contributions

**Conceptualization:** Renata Fiedler Lopes, António José Figueiredo, Carlos Gonçalves, Antonio Tessitore, Ana Maria Teixeira, Luis Rama.

**Data curation:** Renata Fiedler Lopes, Luciele Guerra Minuzzi, Antonio Tessitore, Laura Capranica, Ana Maria Teixeira, Luis Rama.

**Formal analysis:** Renata Fiedler Lopes, Luis Rama.

**Funding acquisition:** António José Figueiredo, Luis Rama.

**Investigation:** Renata Fiedler Lopes.

**Methodology:** Renata Fiedler Lopes, Luis Rama.

**Supervision:** Luis Rama.

**Writing – original draft:** Renata Fiedler Lopes, Luciele Guerra Minuzzi, Ana Maria Teixeira, Luis Rama.

**Writing – review & editing:** Renata Fiedler Lopes, Luciele Guerra Minuzzi, António José Figueiredo, Carlos Gonçalves, Antonio Tessitore, Laura Capranica, Ana Maria Teixeira, Luis Rama.

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
