## [Decision Letter · Decision Letter 0]

16 Apr 2020

PONE-D-20-07349

Upper respiratory symptoms (URS) and salivary responses across a season in youth soccer players: a useful and non-invasive approach for predicting URS susceptibility and occurrence in young athletes

PLOS ONE

Dear Dr. Guerra Minuzzi,

Thank you for submitting your manuscript to PLOS ONE. After careful consideration, we feel that it has merit but does not fully meet PLOS ONE’s publication criteria as it currently stands. Therefore, we invite you to submit a revised version of the manuscript that addresses the points raised during the review process.

All comments from the reviewers should be considered, mainly those regarding to the way results are reported and/or anylised. This comment includes also the general suggestion of showing some results in the abstract. Furthermore, the authors should clearly indicate how and where all data is available, as it is only indicated that "it is available".

We would appreciate receiving your revised manuscript by May 31 2020 11:59PM. To enhance the reproducibility of your results, we recommend that if applicable you deposit your laboratory protocols in protocols.io, where a protocol can be assigned its own identifier (DOI) such that it can be cited independently in the future. For instructions see: http://journals.plos.org/plosone/s/submission-guidelines#loc-laboratory-protocols

We look forward to receiving your revised manuscript.

Kind regards,

Pedro Tauler, Ph.D.

Academic Editor

PLOS ONE

Journal Requirements:

2. In your Methods section, please provide additional information about the participant recruitment method and the demographic details of your participants. Please ensure you have provided sufficient details to replicate the analyses such as: a) the recruitment date range (month and year), b) a description of any inclusion/exclusion criteria that were applied to participant recruitment, c) a table of relevant demographic details, d) a statement as to whether your sample can be considered representative of a larger population, e) a description of how participants were recruited, and f) descriptions of where participants were recruited and where the research took place.

Reviewers' comments:

Reviewer's Responses to Questions

**Comments to the Author**

1. Is the manuscript technically sound, and do the data support the conclusions?

Reviewer #1: Partly

Reviewer #2: Yes

Reviewer #3: Yes

2. Has the statistical analysis been performed appropriately and rigorously? 

Reviewer #1: Yes

Reviewer #2: No

Reviewer #3: Yes

3. Have the authors made all data underlying the findings in their manuscript fully available?

Reviewer #1: Yes

Reviewer #2: No

Reviewer #3: Yes

4. Is the manuscript presented in an intelligible fashion and written in standard English?

Reviewer #1: Yes

Reviewer #2: Yes

Reviewer #3: Yes

5. Review Comments to the Author

Reviewer #1: The study presented by Lopes et al. followed a substantial number of young athletes (n=57 players aged U15 to U19) over the course of a competitive season in order to characterize changes in salivary markers of immunity and hormonal changes. They hypothesized that athletes would exhibit different humoral and immune patterns throughout the season depending on their age. Furthermore, they hypothesized that these age-specific differences would be associated with differences in incidences of Upper Respiratory Symptoms (URS). This interesting premise is based on existing literature, showing associations between salivary markers of immunity and rate of URS in young athletes (Moraes et al. 2017), adults (Fahlman et al. 2017; Guilhem et al. 2015) and elderly (Akimoto et al. 2003). The authors report differences in salivary hormonal profile, along with markers of immunity between the age-groups and throughout the season. However, the authors suggest that the measured markers of oral immunity and hormonal status could be used to predict URS in young athletes, despite finding very weak correlations between URS incidence and their outcome measures (r<0.200 for all). The authors should be careful in making bold statements that are not supported by their data. In addition, the data presented does not appear to be very novel.

Major Comments:

1. Associations between salivary markers of immunity and URS presented by the authors are very weak. Both appear fairly stable over time, and correlation coefficients are well below the “r=0.300” limit for “weak” associations accepted in the literature. The authors should not extrapolate their results in suggesting it could be used to predict URS. Additionally, these results should be graphically depicted to improve the overall impact of the paper.

2. Many researchers have argued that decrease in sIgA and sAA concentrations may not be associated with URS, until they reach a critically low value. Per example, Neville et al. discusses the use of a 40% reduction in the athlete’s normal sIgA concentration as a threshold under which greater incidences of URS are observed. The authors describe sIgA and sAA as continuous variables, however did they also assess whether athletes reporting incidences of URS had lower than normal values of sIgA or sAA? Categorizing sAA and sIgA concentrations based on tertiles or quartiles may be strengthen their argument.

3. No training intensity was available for this study, however one could hypothesize that different positions (GK vs defender vs striker etc…) may exhibit differences in the outcome measures. Did the author investigate this?

4. The authors discuss the use of Testosterone/Cortisol ratio as a proxy measure of anabolic/catabolic state in the athletes. The authors collected saliva samples in the evening (18:00-19:00), where salivary cortisol was likely to be low, and thus this ratio would be more influenced by variations in sT. Furthermore sT is known to be negatively associated with overtraining (Maso et al. 2004), but it is not clear whether the authors saw a similar associations.

5. The result section would benefit from having more graphical depiction of the collected data. Per example, it is difficult to judge whether the changes in URS throughout the season was associated with training volume without a figure (see comment 1).

Minor Comments:

1. The authors describe that participants refrained from Caffeine and Food product; however no mention was made about alcohol consumption. Considering the differences in age amongst the participants, could this explain some of the changes seen in salivary Cortisol and Testosterone in the U19 group?

2. Line 308: This sentence should be reworded

3. Figure 1. Amylase is misspelled.

4. Figure 1. Time effect is not represented

Reviewer #2: General:

This is an interesting paper of potential value to the field. Strengths include the long-term monitoring period (whole soccer season), and high level (national division) participants. Weaknesses include infrequent sampling time-points reducing sensitivity to observe changes across time.

Specific:

Abstract: recommend including some actual quantitative data/results (for main/key outcomes) in abstract results section.

L78-87: Suggest change to order or mentioning these points. i.e. start with total number recruited, then number lost (dropout etc), then final number analysed.

Methods general (and subsequent results interpretation): salivary Cortisol and Testosterone are very susceptible to inaccuracy because of blood contamination in samples. It is recommended that this is screened for using biomarker assays (e.g. transferrin) because this cannot be detected visually.

L182-187: I do not fully understand the value this metric? If you want to normalise for comparison, why not express as % of mean monthly load? e.g. October volume was 113% of the season average?

L194: is % of total URS by age-group a fair and meaningful comparison? it would seem more appropriate to report number of episodes as % of team... or more accurately, average number of episodes per player in each age group? And also the % of players experiencing at least 1 episode and/or those experiencing no episodes during the monitoring period/each time in each age group might be useful?

L337-338: or could it be a marker of overtraining or related to accumulated physiological stress, since OT is sometimes associated with a blunted HPA response?

L394: data not presented in results?? How does this compare to other age groups (i.e. what was the % in others) and other times of season? Any stats/sig differences?

L395: what about in January- in line with these points on high prevalence at this time in population?

L396: affection?

L413: direct stimulating effect is too strong an assertion to make based on this design and these data... an association was observed is not the same as definite cause-effect relationship. Need to tone down language and inference here.

L418-419: but does need to be balanced against the desire to stimulate maximal physiological adaptation response?

L431: Then why not report the power (or Effect Size) statistics for your analyses in the results? You do mention ES in methods- this should be reported- and this will tell you if the power is OK or not.

Minor typos:

L118: ...prior to saliva collection...

L121 3 min?

L165: Descriptive statistics

L196: "There were found a significant inverse association...." would read better as "A significant inverse association was found...."

L198: that there was a trend?

L205: this sentence does not make sense in current form (maybe "show the pattern across time for the physiological parameters" or something similar would be better?)?

L260: significantly?

L308: these data are in line with…

L320: observed pattern?

L322: the same as was observed?

L326: Northern

L333: as a potential

L362: or were not selected

L379: make players more likely to experience

L389: possibly because of

L402: out of

L427: acknowledged rather than assumed?

Reviewer #3: Manuscript Number: PONE-D-20-07349

Title: Upper respiratory symptoms (URS) and salivary responses across a season in youth soccer players: a useful and non-invasive approach for predicting URS susceptibility and occurrence in young athletes

Reviewer' Major Comments:

In this manuscript the authors aimed to examine the effect of a competitive season on salivary responses [cortisol (sC), testosterone (sT), Testosterone/Cortisol ratio (sT/C), Immunoglobulin A (sIgA), sIgA secretion rate (srIgA), alpha-amylase (sAA)] and upper respiratory symptoms (URS) occurrence in three teams of male soccer players (Under-15, Under-17 and Under-19 yrs.). The results show No differences were found for monthly training volume between teams. Incidence of URS was higher for the U15. Higher sT and srIgA were observed for the U19, lower sC were found for the U17 and sAA showed higher values for the U15 throughout the season. Monthly variations showed a decrease in sT in August compared to October and November for the U15. The U19 presented an increase in sC in March compared to February, sT/C were higher in March compared to February and April and sAA increased in March compared to September. Negative correlations, controlled for age group, were found between URS occurrence and srIgA (r=-0.190, p=0,001), sAA (r=-0.175, p=0.001) and sT (r=-0.115, p=0.036). The current study confirmed that youth soccer players show different age-related adaptive hormonal and immune responses to training and competition. The results show also that there is an association between training load (volume), mucosal immunity, hormone levels and URS.

The manuscript is well written and provides some new findings on this area. However, several issues must require attention.

In the abstract please give some significant results (e.g. values).

In the introduction some paragraphs must address what and why we need to know this and what the question you are addressing is and is it really that important.

In the introduction, first paragraph: which sport are the authors talking about ? soccer ?

Methods:

The authors have provided a detailed, yet concise, account of their study methods and experimental design. However, some precisions must be addressed:

- Please give rationality for the n values (number of participants). How did you calculate that (power analysis)?

The discussion reflect what the authors found, how it relates to the literature. However there is a lack of what it means physiologically and there are some speculations.

The limitations of the study may state also the cost of such follow-up and the lack of nutritional measurements.

6. PLOS authors have the option to publish the peer review history of their article (what does this mean?). If published, this will include your full peer review and any attached files.

Reviewer #1: No

Reviewer #2: No

Reviewer #3: Yes: Prof. H. ZOUHAL

---

## [Author Response · Author response to Decision Letter 0]

3 Jun 2020

Reviewer 1: I have incorporate all of your suggestions into my review. They were very helpful. Thank you.

Reviewer 2: I have incorporate all of your suggestions into my review. They were very helpful. Thank you.

Reviewer 3: I have incorporate all of your suggestions into my review. They were very helpful. Thank you.

---

## [Decision Letter · Decision Letter 1]

19 Jun 2020

PONE-D-20-07349R1

Upper respiratory symptoms (URS) and salivary responses across a season in youth soccer players: a useful and non-invasive approach associated to URS susceptibility and occurrence in young athletes

PLOS ONE

Dear Dr. Guerra Minuzzi,

Thank you for submitting your manuscript to PLOS ONE. After careful consideration, we feel that it has merit but does not fully meet PLOS ONE’s publication criteria as it currently stands. Therefore, we invite you to submit a revised version of the manuscript that addresses the points raised during the review process by reviewer 2 about previous comments not completely clarified or revised by the authors.

We look forward to receiving your revised manuscript.

Kind regards,

Pedro Tauler, Ph.D.

Academic Editor

PLOS ONE

Reviewers' comments:

Reviewer's Responses to Questions

**Comments to the Author**

1. If the authors have adequately addressed your comments raised in a previous round of review and you feel that this manuscript is now acceptable for publication, you may indicate that here to bypass the “Comments to the Author” section, enter your conflict of interest statement in the “Confidential to Editor” section, and submit your "Accept" recommendation.

Reviewer #1: All comments have been addressed

Reviewer #2: (No Response)

Reviewer #3: All comments have been addressed

2. Is the manuscript technically sound, and do the data support the conclusions?

Reviewer #1: Yes

Reviewer #2: Yes

Reviewer #3: Yes

3. Has the statistical analysis been performed appropriately and rigorously? 

Reviewer #1: Yes

Reviewer #2: Yes

Reviewer #3: Yes

4. Have the authors made all data underlying the findings in their manuscript fully available?

Reviewer #1: Yes

Reviewer #2: Yes

Reviewer #3: Yes

5. Is the manuscript presented in an intelligible fashion and written in standard English?

Reviewer #1: Yes

Reviewer #2: Yes

Reviewer #3: Yes

6. Review Comments to the Author

Reviewer #1: The authors have appropriately responded to my concerns. The updated manuscript provides novel insight on the immune function of youth soccer players

Reviewer #2: Thank you for addressing most of my comments. I have the following comments that require further consideration:

Regarding the following comment and author response: Abstract: recommend including some actual quantitative data/results (for main/key outcomes) in abstract results section.

Author response 1: We agreed with the recommendation. However, we opted for a descriptive presentation of the results in the abstract, as we think it would be the clearest way.

FURTHER COMMENT ON THIS:

This reviewer does not agree with this response. The current presentation is not informative to readers. P values and correlations are of little use or meaning without some actual data to benchmark this against. It is essential to include some data (not for all, but for at least THE most important measure).

Regarding the following comment and author response: Methods general (and subsequent results interpretation): salivary Cortisol and Testosterone are very susceptible to inaccuracy because of blood contamination in samples. It is recommended that this is screened for using biomarker assays (e.g. transferrin) because this cannot be detected visually.

Author response 1: Thanks for the remark. All subjects in the study were checked on a regular base for oral health by the academy medical staff. We take every care concerning to all Salimetrics indications for saliva sampling. The variation of hormonal and immune results is under expected values and in the range of the commercial kits used for analyses. When detected, some values out of the range, were excluded from analysis. Although we did not use any biomarker for blood contamination, we believe that we met all care possible.

FURTHER COMMENT ON THIS:

“All of the Salimetrics indications”? This is not completely clear, but blood ‘contamination’ is quite normal and not something that can be determined only by oral health per se (so the regular check-ups are not sufficient to protect against this. Many players could be perfectly healthy and normal in terms of oral check-ups etc, but can still provide samples with blood contamination sufficient to affect accuracy of T and C measured in these samples). Importantly, this can vary across the day and day-to-day within the same person. You will not necessarily be able to pick this up by whether or not the values are in the normal range, it could still be present and could still skew the results of a particular sample. In summary, you cannot be certain if blood contamination was present or not in each sample with the methods you have employed and there remains the possibility that this could have caused some inaccuracies in the results you report. This must be noted and acknowledged in the limitations section (and hence be careful with any conclusions that you draw based on these particular results).

Regarding the following comment and author response: L182-187: I do not fully understand the value this metric? If you want to normalise for comparison, why not express as % of mean monthly load? e.g. October volume was 113% of the season average?

Author response 1: We thank the reviewerfor the opportunity to clarify the presentation of the seasonal variation in the training volume observed in the 3 teams. We calculate the mean seasonal volumes(training and match)and the monthly variation around this value(Figure 1). Regarding U15 group we observed that the highest value was in October (12.3% above), whereas the U17 and U19 showed higher volumes in August, respectively 18,6% and 14,5% above de seasonal mean volume.The seasonal variation of training was low, for both teams (CV= 9,7%, 10,3% and 11,3%) for U-15, U17 and U-19, respectively.

FURTHER COMMENT ON THIS:

Thanks for the clarification. This is not completely clear in the accompanying text however (i.e. L203-205 of the revised manuscript). I would recommend adding some clarification (i.e. to better explain that the 12.3% represents the % increase above average… so perhaps it would be clearer as follows (or something similar)?

“Regarding the monthly distribution of exercise volume in each group, the U15 highest percentages emerged in October (12.3% above season average), whereas the relative picture for both U17 (18.6% above season average) and U19 (14.5% above season average) was in August.”

Reviewer #3: Please use the space provided to explain your answers to the questions above. You may also include additional comments for the author, including concerns about dual publication, research ethics, or publication ethics. (Please upload your review as an attachment if it exceeds 20,000 characters) (Limit 100 to 20000 Characters)

The authors respond to all our comments and suggestions. Congratulations.

7. PLOS authors have the option to publish the peer review history of their article (what does this mean?). If published, this will include your full peer review and any attached files.

Reviewer #1: No

Reviewer #2: No

Reviewer #3: Yes: Prof. Hassane ZOUHAL

---

## [Author Response · Author response to Decision Letter 1]

9 Jul 2020

Dear,

We would like to thank the reviewers and editor for their new careful review of the manuscript. They raise another important issue and their inputs are very helpful for improving the manuscript. We now submit a new revised version of the article. We hope that the reviewers and editor will find the new version of the manuscript satisfactory.

 Here, we present a response to each point raised by the reviewer 2. Also, we clarify the text and the changes are in red in the manuscript version upload. 

6. Review Comments to the Author

Reviewer #1: The authors have appropriately responded to my concerns. The updated manuscript provides novel insight on the immune function of youth soccer players

Reviewer #2: Thank you for addressing most of my comments. I have the following comments that require further consideration:

Regarding the following comment and author response: Abstract: recommend including some actual quantitative data/results (for main/key outcomes) in abstract results section.

Author response 1: We agreed with the recommendation. However, we opted for a descriptive presentation of the results in the abstract, as we think it would be the clearest way.

FURTHER COMMENT ON THIS:

This reviewer does not agree with this response. The current presentation is not informative to readers. P values and correlations are of little use or meaning without some actual data to benchmark this against. It is essential to include some data (not for all, but for at least THE most important measure).

Author response 2: Thanks for the remark and the opportunity to improve this section. We hope that we understand well your vision, and we added the most critical data in the abstract, given it more informative to the readers! We choose to use the percental variation of hormonal and immune parameters values during the season. But, if necessaire, we are open to change and also present the absolute values of the data.

Regarding the following comment and author response: Methods general (and subsequent results interpretation): salivary Cortisol and Testosterone are very susceptible to inaccuracy because of blood contamination in samples. It is recommended that this is screened for using biomarker assays (e.g. transferrin) because this cannot be detected visually.

Author response 1: Thanks for the remark. All subjects in the study were checked on a regular base for oral health by the academy medical staff. We take every care concerning to all Salimetrics indications for saliva sampling. The variation of hormonal and immune results is under expected values and in the range of the commercial kits used for analyses. When detected, some values out of the range, were excluded from analysis. Although we did not use any biomarker for blood contamination, we believe that we met all care possible.

FURTHER COMMENT ON THIS:

“All of the Salimetrics indications”? This is not completely clear, but blood ‘contamination’ is quite normal and not something that can be determined only by oral health per se (so the regular check-ups are not sufficient to protect against this. Many players could be perfectly healthy and normal in terms of oral check-ups etc, but can still provide samples with blood contamination sufficient to affect accuracy of T and C measured in these samples). Importantly, this can vary across the day and day-to-day within the same person. You will not necessarily be able to pick this up by whether or not the values are in the normal range, it could still be present and could still skew the results of a particular sample. In summary, you cannot be certain if blood contamination was present or not in each sample with the methods you have employed and there remains the possibility that this could have caused some inaccuracies in the results you report. This must be noted and acknowledged in the limitations section (and hence be careful with any conclusions that you draw based on these particular results).

Author response 2: We thank the opportunity, and we followed the reviewer advice and suggestion. We added this information clarifying that the samples have not been screened for possible blood contamination which is a limitation of the study.

Regarding the following comment and author response: L182-187: I do not fully understand the value this metric? If you want to normalise for comparison, why not express as % of mean monthly load? e.g. October volume was 113% of the season average?

Author response 1: We thank the reviewer for the opportunity to clarify the presentation of the seasonal variation in the training volume observed in the 3 teams. We calculate the mean seasonal volumes(training and match)and the monthly variation around this value(Figure 1). Regarding U15 group we observed that the highest value was in October (12.3% above), whereas the U17 and U19 showed higher volumes in August, respectively 18,6% and 14,5% above de seasonal mean volume.The seasonal variation of training was low, for both teams (CV= 9,7%, 10,3% and 11,3%) for U-15, U17 and U-19, respectively.

FURTHER COMMENT ON THIS:

Thanks for the clarification. This is not completely clear in the accompanying text however (i.e. L203-205 of the revised manuscript). I would recommend adding some clarification (i.e. to better explain that the 12.3% represents the % increase above average… so perhaps it would be clearer as follows (or something similar)?

“Regarding the monthly distribution of exercise volume in each group, the U15 highest percentages emerged in October (12.3% above season average), whereas the relative picture for both U17 (18.6% above season average) and U19 (14.5% above season average) was in August.”

Author response 2: We agree and thanks the reviewer. The correction in the text was done following the reviewer advice.

Reviewer #3: Please use the space provided to explain your answers to the questions above. You may also include additional comments for the author, including concerns about dual publication, research ethics, or publication ethics. (Please upload your review as an attachment if it exceeds 20,000 characters) (Limit 100 to 20000 Characters)

The authors respond to all our comments and suggestions. Congratulations.

---

## [Decision Letter · Decision Letter 2]

13 Jul 2020

Upper respiratory symptoms (URS) and salivary responses across a season in youth soccer players: a useful and non-invasive approach associated to URS susceptibility and occurrence in young athletes

PONE-D-20-07349R2

Dear Dr. Guerra Minuzzi,

We’re pleased to inform you that your manuscript has been judged scientifically suitable for publication and will be formally accepted for publication once it meets all outstanding technical requirements.

Kind regards,

Pedro Tauler, Ph.D.

Academic Editor

PLOS ONE

Additional Editor Comments (optional):

Reviewers' comments:

Reviewer's Responses to Questions

**Comments to the Author**

1. If the authors have adequately addressed your comments raised in a previous round of review and you feel that this manuscript is now acceptable for publication, you may indicate that here to bypass the “Comments to the Author” section, enter your conflict of interest statement in the “Confidential to Editor” section, and submit your "Accept" recommendation.

Reviewer #2: All comments have been addressed

2. Is the manuscript technically sound, and do the data support the conclusions?

Reviewer #2: Yes

3. Has the statistical analysis been performed appropriately and rigorously? 

Reviewer #2: Yes

4. Have the authors made all data underlying the findings in their manuscript fully available?

Reviewer #2: (No Response)

5. Is the manuscript presented in an intelligible fashion and written in standard English?

Reviewer #2: Yes

6. Review Comments to the Author

Reviewer #2: Thanks

7. PLOS authors have the option to publish the peer review history of their article (what does this mean?). If published, this will include your full peer review and any attached files.

Reviewer #2: No

---

## [Editor Report · Acceptance letter]

24 Jul 2020

PONE-D-20-07349R2 

Upper respiratory symptoms (URS) and salivary responses across a season in youth soccer players: a useful and non-invasive approach associated to URS susceptibility and occurrence in young athletes 

Dear Dr. Minuzzi:

I'm pleased to inform you that your manuscript has been deemed suitable for publication in PLOS ONE. Congratulations! Your manuscript is now with our production department. 

Kind regards, 

on behalf of

Dr. Pedro Tauler 

Academic Editor

PLOS ONE